# A robust method for particulate detection of a genetic tag for 3D electron microscopy

James Rae[1], Charles Ferguson[1], Nicholas Ariotti[2,3], Richard I Webb[4], Han-Hao Cheng[4], James L Mead[4,5], James D Riches[6], Dominic JB Hunter[1,7], Nick Martel[1], Joanne Baltos[8], Arthur Christopoulos[8], Nicole S Bryce[3], Maria Lastra Cagigas[3], Sachini Fonseka[1], Marcel E Sayre[4], Edna C Hardeman[3], Peter W Gunning[3], Yann Gambin[7], Thomas E Hall[1], Robert G Parton[1,4]*

[1]The University of Queensland, Institute for Molecular Bioscience, Queensland, Australia; [2]Mark Wainwright Analytical Centre, University of New South Wales, Sydney, Australia; [3]School of Medical Sciences, University of New South Wales, Sydney, Australia; [4]The University of Queensland, Centre for Microscopy and Microanalysis, Queensland, Australia; [5]Division Microrobotics and Control Engineering, Department of Computing Science, University of Oldenburg, Oldenburg, Germany; [6]Queensland University of Technology, Queensland, Australia; [7]EMBL Australia Node for Single Molecule Sciences, University of New South Wales, Sydney, Australia; [8]Monash Institute of Pharmaceutical Sciences, Monash University, Victoria, Australia

*For correspondence:
r.parton@imb.uq.edu.au

Competing interests: The authors declare that no competing interests exist.

**Abstract** Genetic tags allow rapid localization of tagged proteins in cells and tissues. APEX, an ascorbate peroxidase, has proven to be one of the most versatile and robust genetic tags for ultrastructural localization by electron microscopy (EM). Here, we describe a simple method, APEX-Gold, which converts the diffuse oxidized diaminobenzidine reaction product of APEX into a silver/gold particle akin to that used for immunogold labelling. The method increases the signal-to-noise ratio for EM detection, providing unambiguous detection of the tagged protein, and creates a readily quantifiable particulate signal. We demonstrate the wide applicability of this method for detection of membrane proteins, cytoplasmic proteins, and cytoskeletal proteins. The method can be combined with different EM techniques including fast freezing and freeze substitution, focussed ion beam scanning EM, and electron tomography. Quantitation of expressed APEX-fusion proteins is achievable using membrane vesicles generated by a cell-free expression system. These membrane vesicles possess a defined quantum of signal, which can act as an internal standard for determination of the absolute density of expressed APEX-fusion proteins. Detection of fusion proteins expressed at low levels in cells from CRISPR-edited mice demonstrates the high sensitivity of the APEX-Gold method.

## Introduction

Genetic tags for electron microscopy (EM) have made protein ultrastructural localization possible within the three-dimensional (3D) environment of cells, tissues, and whole organisms (*Ariotti et al., 2015*; *Han et al., 2019*; *Lam et al., 2015*; *Martell et al., 2017*; *Martell et al., 2012*; *Tsang et al., 2018*). APEX, a modified ascorbate peroxidase derived from soybean, is a simple highly versatile marker for EM detection. The wide applicability of its use is demonstrated by studies utilizing APEX and its derivative, APEX2, for detection of fusion proteins in cells, *Drosophila*, zebrafish, and mice

(*Ariotti et al., 2015*; *Hirabayashi et al., 2018*; *Li et al., 2020*; *Meiring et al., 2019*). APEX has also been combined with nanobody-based detection systems for rapid localization of GFP- and mCherry-tagged proteins (*Ariotti et al., 2015*; *Ariotti et al., 2018*), and used to detect protein interactions using split GFP and nanobodies or using a novel split-APEX system (*Ariotti et al., 2018*; *Han et al., 2019*). These applications are compatible with 3D EM techniques in which the reaction product can be produced within the depth of the specimen, rather than on the surface as occurs with labelling on sections (*Martell et al., 2017*; *Martell et al., 2012*), and can also be used with methods that require cryo-preservation (*Tsang et al., 2018*). Despite its versatility, the relatively diffuse diaminobenzidine (DAB) reaction product can make distinguishing the APEX reaction from electron-dense cellular components difficult and often requires an expert to interpret the images. This is especially apparent when APEX-tagged proteins have multiple subcellular localizations, for example, a soluble and a membrane-localized distribution (*Follett et al., 2016*). A method which allows researchers to obtain a particulate signal from a genetic tag that could be equated to the actual number of antigens present would be a huge breakthrough in the field.

In this study, we describe the use of a method for converting the DAB reaction produced by a cytoplasmically exposed APEX tag into a particulate marker. This method produces an easily detectable gold particle at the site of the fusion protein with high resolution and specificity, allowing simple correlative light and electron microscopy (CLEM), use in 3D EM techniques, and simple quantitation.

## Design

In order to convert the DAB reaction product to a particulate marker, we tested a number of protocols with a particular focus on silver/gold (Ag/Au) enhancement methods originally developed for amplification of the DAB signal obtained with peroxidase-labelled antibodies on histological sections. The criteria for enhancement of the APEX-DAB reaction product for EM were: (1) production of a uniformly sized particle with high specificity, (2) high sensitivity with low background and no self-nucleation, (3) high resolution, and (4) ease of use, that is, using conventional fixation and processing schemes, readily available laboratory reagents, and in ambient light conditions rather than a darkroom. The optimal protocol which satisfied these criteria was a modified Ag/Au enhancement method (*Sedmak et al., 2009*) as shown schematically in *Figure 1A*, similar to that used to visualize luminal APEX (*Mavlyutov et al., 2017*). After fixation and a conventional incubation with DAB/$H_2O_2$ to reveal the oxidized DAB reaction product, cells were incubated sequentially with a silver nitrate solution (containing hexamethylenetetramine and disodium tetraborate) and then with gold chloride. This resulted in local production of stabilized gold particles in the range of 10–15 nm in diameter as the argyrophilic oxidized DAB reaction product converts the metal salts to colloidal particles at the site of the fusion protein. Gum arabic was included to provide consistent uniform nucleation.

## Results

We applied this localization method to Cavin4-APEX2 which is associated with cell surface caveolae (*Figure 1B,D,E* and *Figure 1—figure supplement 1A,B*), to LifeAct-APEX2 that allows ultrastructural detection of actin filaments (*Figure 1C* and *Figure 1—figure supplement 1C,D*), and to A1AR-APEX2, a G-protein-coupled receptor (*Figure 1—figure supplement 2D*). The APEX-Gold method satisfies the criteria of specificity, low background, sensitivity, resolution, and ease of use. Untransfected cells show negligible Ag/Au particles (*Figure 1—figure supplement 2A*) and labelling is tightly restricted to caveolae with an average of 18.4 nm from the caveolar membrane to the centre of the particulate reaction product when imaged using transmission EM (*Figure 1—figure supplement 2B,C*). The experimental process is very simple and robust; all incubations are done in the light and require no specialist chemicals or equipment. The method has been used successfully in over 20 different biological replicate experiments using the Cavin4-APEX2 and LifeAct-APEX2 systems. Critically, it results in an unambiguous particulate reaction product that is clearly definable without expert interpretation. Sections were generally viewed without further on-grid staining to maximize detection of gold particles. However, visualization of the electron-dense APEX-Gold particles is readily compatible with on-grid staining (*Figure 1—figure supplement 2A*). We were also able to visualize APEX-Gold labelling with Tokuyasu-cryosectioned samples (*Figure 1—figure supplement 3A–C*).

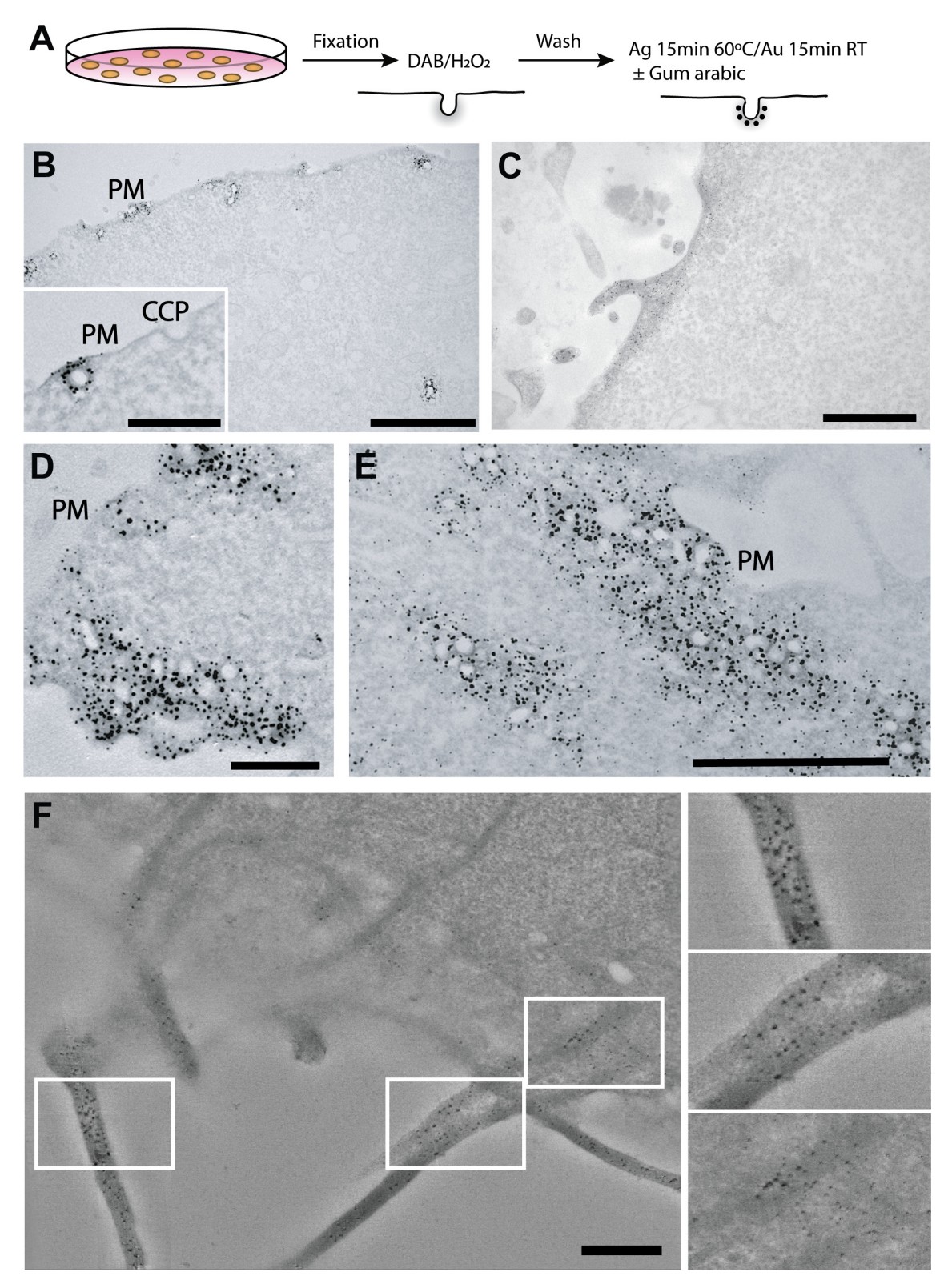

**Figure 1.** APEX-Gold particulate labelling of genetically-tagged proteins of interest. (**A**) Schematic of the APEX-Gold method. Cells were transfected with Cavin4-APEX2 (**B,D,E**) (control light microscopy experiments are described in *Figure 1—figure supplement 5*) or LifeAct-APEX2 (**C,F**), fixed, treated with diaminobenzidine (DAB), and then incubated with Ag/Au reagents in the presence of gum arabic. (**B,D,E**) Low (**B**) and higher (**D,E**, inset in B) magnification views of caveolae labelling. (**C**) Labelled actin filaments. (**F**) Optical slice projection through tomogram of LifeAct-APEX2 expressing

*Figure 1 continued on next page*

*Figure 1 continued*

cells. APEX-Gold particulate reaction product can be observed tightly associated with and throughout the actin bundles in three dimensions. Note the uniform gold label, the lack of background, and high signal to noise. PM, plasma membrane; CCP, clathrin-coated pit. Bars, B, 2 µm (inset 500 nm); C, 1 µm; D, 500 nm; E, 1 µm; F, 500 nm.

The online version of this article includes the following figure supplement(s) for figure 1:

**Figure supplement 1.** Cavin-4 and LifeAct APEX-Gold labelling.
**Figure supplement 2.** APEX-Gold produces an easy to identify signal with little background, high specificity and broad applicability.
**Figure supplement 3.** Cryo-sectioning and SEM array tomography of APEX-Gold-labelled cells.
**Figure supplement 4.** APEX-Gold is compatible with serial block-face SEM, FIB-SEM, image segmentation analysis and pre-embedding labelling techniques.
**Figure supplement 5.** Brightfield images of BHK cells with various treatments.

The clarity and density of the APEX-Gold enhanced particulate signal makes the method compatible with 3D EM methods including 3D electron tomography (*Figure 1F*), array tomography scanning EM (SEM) (*Figure 1—figure supplement 3D,E*), focussed ion beam (FIB) EM (*Figure 1—figure supplement 4B*), and serial block-face SEM (*Figure 1—figure supplement 4A*). The gold particles can also be resolved by supervised and automated segmentation (Weka Image J/FIJI plug-in, *Figure 1—figure supplement 4B',C'*). We also tested the compatibility of the method with freeze substitution/low temperature embedding (*Figure 1—figure supplement 4D,E*). This makes APEX-Gold potentially compatible with double labelling, with the APEX-fusion protein being expressed endogenously in the cells and then sections labelled for other proteins of interest. The APEX-Gold method is also simple to use in CLEM approaches (*Figure 2C–F*) and can be checked by dot blot in parallel to the EM experiment to ensure the protocol is standardized (*Figure 2—figure supplement 1B*; see Materials and methods for a standard protocol).

Next, we investigated whether we could develop a system to act as an internal control for the APEX-Gold method and to allow quantitative comparison with cellular APEX-tagged proteins of interest. We made use of the ability of mammalian caveolin-1 (CAV1), the major structural protein of caveolae, to generate nanovesicles with a defined number of caveolin proteins when expressed in a cell-free system (*Jung et al., 2018*) (see scheme in *Figure 2A*). CAV1-APEX2 was expressed in a cell-free Leishmania lysate (GFP-tagged and -untagged; *Figure 2—figure supplement 1C*) and the GFP-tagged protein characterized by fluorescence correlation spectroscopy (FCS) (*Figure 2—figure supplement 1A*). The resulting vesicles contained a quantum of fluorescence consistent with approximately 110 GFP-CAV1-APEX2 molecules per vesicle. We then compared the enzymatic activity of the cell-free synthesized GFP-CAV1-APEX-fusion protein with commercial horseradish peroxidase using dot blots. The cell-free synthesized APEX2 fusion protein had higher activity per µg of protein than the commercial HRP preparation (*Figure 2—figure supplement 1B*). Enhancement of the signal using the APEX-Gold protocol caused a slight increase in sensitivity of detection and a colour change in the dot blot providing a simple assay to check for successful enhancement.

We next used the uniformly sized in vitro generated GFP-CAV1-APEX2 vesicles for comparative studies in cells. The lysate containing the GFP-CAV1-APEX2 vesicles was added to cultured A431 cells for 5 min at 37˚C prior to fixation, DAB treatment, and APEX-Gold enhancement. As shown in *Figure 2B*, the GFP-CAV1-APEX2 vesicles were observed on the surface of the cells and in endosomes and are clearly decorated with the APEX-Gold enhanced particles (*Figure 2—figure supplement 1D*). Under these enhancement conditions, an average of 16.9 Ag/Au particles per vesicle (mean ± 4.3 SD, n = 26) was observed in the approximately 100-nm-thick section. This value could be used as a standard to examine the efficiency of APEX-Gold particulate formation, when coupled with the previously established molecules per vesicle (~110). This proof of principle experiment using exogenously added Caveolin-APEX vesicles as a standard demonstrates the potential of the APEX-Gold method for determination of the density of unknown proteins.

While these experiments illustrated the potential of APEX-Gold for quantitative studies, it was apparent from close examination of labelled CAV1-APEX that separating individual nucleation events for the densely packed APEX2-tags was problematic. We observed a varied distribution of gold sizes which we hypothesize are a consequence of fusion of multiple individual particulates into a single APEX-Gold product (*Figure 2—figure supplement 1D*). We speculated that a low abundance antigen might be detected with high efficiency if the APEX2-tagged proteins are well spaced.

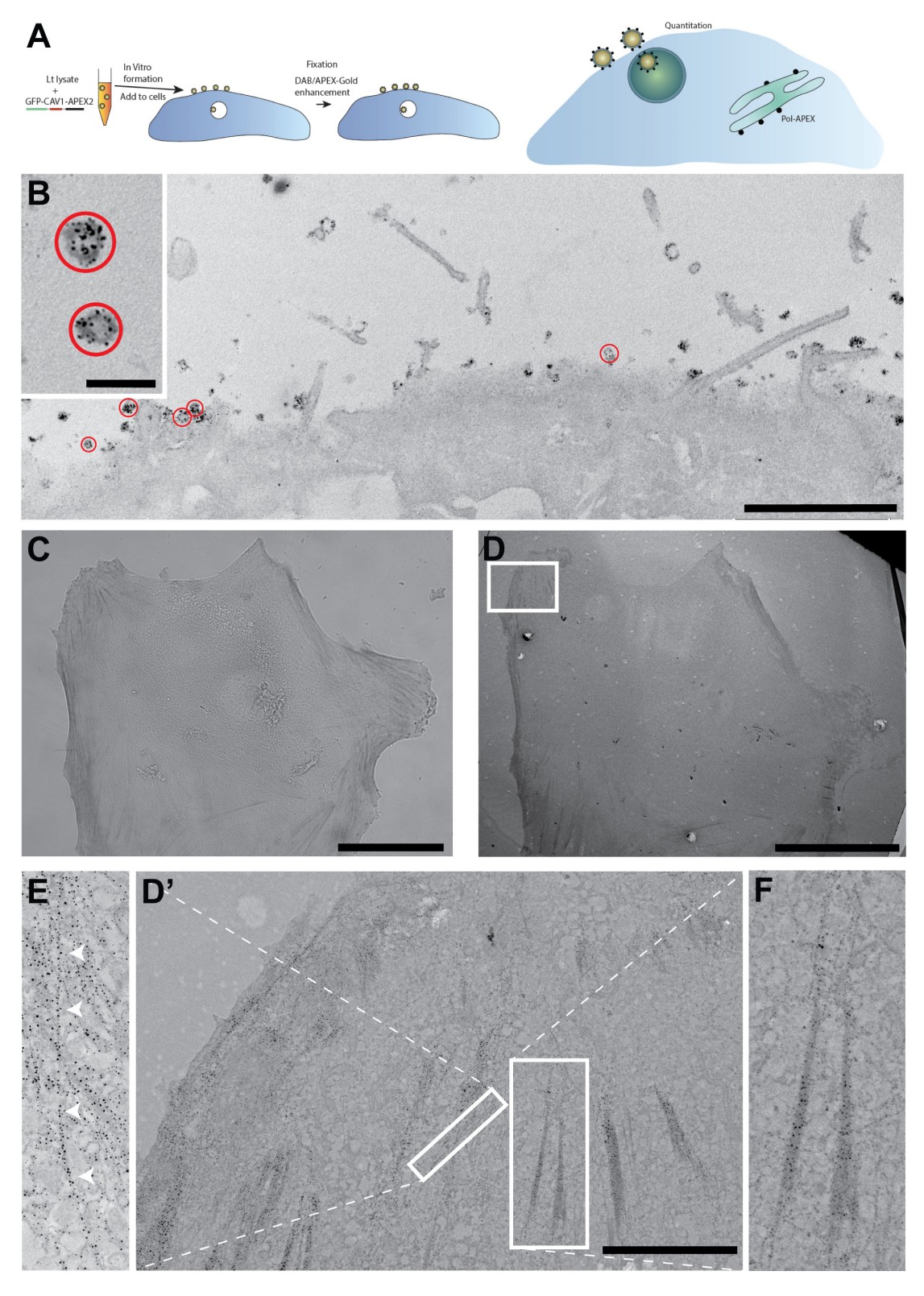

**Figure 2.** APEX-fusion protein density determination using an internal standard and low level protein detection. (A) Schematic explaining the cell-free caveolae-APEX2-Gold system. (B) A431 cells were incubated with in vitro synthesized CAV1-APEX2 cell-free caveolae for 5 min at 37°C before fixation and processing for APEX-Gold detection. Note the Ag/Au labelling of the surface-associated cell-free caveolae circled in red and low background label within cell. (C) Light microscopic detection of tropomyosin 3.1 (Tpm3.1)-APEX2 after APEX-Gold DAB/Ag/Au detection. (D) Low magnification electron

*Figure 2 continued on next page*

*Figure 2 continued*

microscopy (EM) showing a basal section of the same cell. (D',F) Higher magnification views of gold labelled stress fibres from boxed region. (D',E) Gold labelling follows individual actin filaments from boxed region. Bars, B, 2 μm (inset 200 nm); C,D, 50 μm; D', 5 μm.

The online version of this article includes the following figure supplement(s) for figure 2:

**Figure supplement 1.** Cell free GFP-Cav1-Apex2 vesicle characterisation.
**Figure supplement 2.** Low abundance Tpm3.1 labelling highlights APEX-Gold sensitivity and resolution.
**Figure supplement 3.** Tpm3.1 APEX-Gold labelling.

To test this we examined an APEX2-tagged tropomyosin isoform. APEX-tagged tropomyosin is a good candidate to obtain precise physical organization because it is an extended molecule (~35 nm; *Sousa et al., 2010*) and so the APEX molecules should be well spaced along the actin filament. We utilized a recently characterized CRISPR-generated mouse line that expresses tropomyosin 3.1 (Tpm3.1) C-terminally tagged with APEX2 from the endogenous locus (*Meiring et al., 2019*). In these mice, exon 9d of the tropomyosin gene is fused to APEX2 via a 10 amino acid linker (*Meiring et al., 2019*; *Figure 2—figure supplement 2A*). Western blotting of mouse embryonic fibroblasts (MEFs) prepared from Tpm3.1-APEX2 heterozygous mice showed that expression of the fusion protein is lower than that of the endogenous Tpm3.1 protein (~5%; *Figure 2—figure supplement 2B,C*). We then subjected the MEFs to the APEX-Gold method for detection of the fusion protein. Light microscopic studies revealed that specific labelling was associated with putative stress fibres (*Figure 2C*), and this was confirmed by correlative EM (*Figure 2D–F*) that showed a dense and highly specific particulate labelling associated with bundles of actin, with an average of 10.6 nm from the particle to the centre of bundle mass when imaged using transmission EM (*Figure 2—figure supplement 2D–F*). In addition, particulate labelling was more uniform in size distribution compared to CAV1-APEX vesicles and followed distinct tracks suggestive of individual actin filaments (*Figure 2E* and *Figure 2—figure supplement 3A–F*). The precise arrangement of tropomyosin on individual actin filaments is not yet known but the gold spacing along actin filaments (*Figure 2E*) demonstrates the sensitivity of the method and the feasibility of this approach. Moreover, the greater uniformity of the gold reaction product in the Tpm.3.1-APEX2 MEFs suggests that the packing or density of tags may impact gold size distribution. These results clearly demonstrate the power of the APEX-Gold technique for detecting proteins at low expression levels and/or proteins expressed from endogenous loci.

## Discussion

Here, we present a simple technique for identification of a protein of interest by transmission EM using a genetic tag to generate a particulate signal. The method has high resolution, low background, and is sensitive enough to detect proteins expressed at or below endogenous levels. We provide a defined standard for calibration to ensure reproducibility and show that the method is compatible with 3D EM techniques such as serial block-face SEM, array tomography, FIB SEM, and electron tomography.

The APEX system has proven to be a powerful new addition to the biologist's arsenal of techniques with wide applications in EM and in proteomics. The original APEX system involving a fusion with the protein of interest has been extended to new modifications including nanobody-based detection of fluorescent proteins and protein complexes and a split-APEX which only yields an active enzyme if brought together by two partner proteins. All of these applications are compatible with the APEX-Gold method for conversion of DAB to an Ag/Au particle. The APEX-Gold method is so simple that it should be used in parallel to light microscopy as an initial step in protein localization. In fact, through the use of nanobody-APEX constructs, a co-transfection with a GFP-tagged protein can yield CLEM images in a very short timeframe. The technique produces a particulate signal which is clearly distinguished from any cellular feature and is readily quantitated. By combining with an APEX-fusion protein of known signal density (here demonstrated by cell-free generated vesicle of a defined APEX density as a calibration standard), it is possible to quantitatively compare particle density to antigen density. While we demonstrated this principle by using exogenously added cell-free synthesized caveolin-AP vesicles, an APEX-tagged antigen generated by CRISPR technology that is present at known density in a specific region of the cell could also be used as a standard.

We were able to use the APEX-Gold method to localize low levels of APEX2-tagged proteins and to estimate the sensitivity of the technique. We utilized cells from mice genetically modified to express APEX2 fused to Tpm3.1 which have been previously characterized in detail (*Meiring et al., 2019*). Embryonic fibroblasts from the mice express the Tpm3.1-APEX2 fusion protein at approximately 5% of endogenous levels. Tpm3.1 is expressed at a concentration of 30 µM (*Meiring et al., 2018*) and so 5% represents a concentration of 1.5 µM. This equates to a cellular Tpm3.1-APEX2 copy number of approximately 1 million Tpm3.1 homodimers per cell with 250,000 predicted to be associated with actin and the remainder cytosolic (*Meiring et al., 2018*). This level of expression resulted in an excellent signal-to-noise ratio for the APEX-Gold particles on actin filaments, facilitated by the spacing of the gold along the filaments which is related to the size and organization of the tropomyosin oligomers. This demonstrates the remarkable efficiency of antigen detection and lays the foundation for truly quantitative immuno-EM in cells and tissues for the first time.

APEX-Gold represents a significant advance in protein distribution analysis by EM. Compared with current APEX methods which rely on subjective interpretation of a diffuse DAB reaction product, the improved signal-to-noise ratio and a particulate, quantitative readout indicate that this method should become the gold standard for localization of genetically tagged proteins in EM.

## Methodological considerations

The method used here, which we have optimized for APEX localization, is based on well-established methods for histological and EM visualization of the insoluble DAB product of peroxidase (*Adams, 1981*; *Danscher and Nörgaard, 1983*; *Dobó et al., 2011*; *Newman and Jasani, 1998*; *Pohl and Stierhof, 1998*), combined with the methods of silver reduction and gold toning developed for photography in the 19th century (for review, see *Ellis, 1975*). These methods have been developed and used in many laboratories to allow enhancement of the DAB reaction for light microscopic applications, for converting the DAB signal to an easily detected particulate marker to allow discrimination from any cellular structures, and to increase the sensitivity of DAB detection (*Dobó et al., 2011*; *Newman et al., 1983*; *Sedmak et al., 2009*). The three principle steps involve: (1) production of the polymerized DAB product, (2) the reduction of silver ions to submicroscopic metallic silver by the argyrophilic DAB polymer, and (3) substitution of metallic silver by gold (gold toning), through the simple reaction $Ag + Au^+ \rightarrow Ag^+ + Au$ to produce the more inert gold particle that can resist subsequent osmium treatment. A further refinement is the use of gum arabic which was introduced for gold enhancement by *Danscher, 1981*; *Danscher and Nörgaard, 1983* to allow precise control of Ag/Au nucleation and more uniform particle generation.

While the benefits of the APEX-Gold method are clear, this approach still comes with certain caveats that must be considered prior to performing these analyses. First, it is necessary to take care to precisely control the development time for particulate deposition with the silver enhancement and subsequent gold toning. We optimized the development conditions to suit the size distribution of the particulate reaction product for detection of APEX-tags at a local density that was suitable for visualization of caveolar-proteins and actin-associated proteins at a reasonable magnification. Increased development times can increase the particle size, facilitating their identification, but can cause appearance of artefactual self-nucleated particles at sites lacking APEX (hence the need for careful control experiments). Increased enhancement times can also cause more variable particle size including the appearance of 'fused' gold particles where individual APEX molecules are closely packed and so not clearly resolved as individual gold particles after enhancement (see for example *Figure 2—figure supplement 1D*). This can cause potential difficulties for quantitation and for applications in double labelling, for example, when the APEX-Gold method is combined with on-section labelling of Tokuyasu sections or freeze-substituted Lowicryl sections. It is also possible that the enhancement might differ when the APEX molecule is present in different cellular compartments (e.g. cytosol versus endosome lumen). However, these issues are dependent on the system being studied; in experiments involving detection of tropomyosin-APEX (*Figure 2C–F*), gold particles were generally well spaced and well resolved as individual particles and of uniform size (*Figure 2—figure supplement 2D*). This suggests that variation is dependent on the density of the antigen of interest. Optimization of the enhancement time is required to allow detection of individual gold particles without formation of single large particles from multiple closely packed APEX molecules. Careful examination of the labelling pattern for Cavin4 reveals larger particles close to the caveolar membrane, and smaller particles in the neighbouring cytosol. This difference in size may be explained by

the existence of Cavin4 in different oligomeric states on the caveolar membrane and in the cytosol (*Kovtun et al., 2014*), that is, it reflects a real biological difference which could be used to infer the state of the protein in particular sites. This problem can potentially be avoided by expressing low levels of APEX-tagged proteins, as shown here for cells expressing an APEX-tagged tropomyosin isoform. We have also only shown proteins directly tagged with APEX2 in this study. However, the system is readily compatible with the use of conditionally stabilized APEX-nanobodies to a GFP- or mCherry-tagged protein of interest as used previously (*Ariotti et al., 2018*). This would affect the ability to correlate the APEX-Gold signal with the density of the protein of interest but has advantages for CLEM approaches or for detection of protein complexes by using split GFP. Despite these caveats, the APEX-Gold method can provide superior signal-to-noise ratios, allows absolute quantitation of particulate signals, is compatible with 3D EM methods and automated segmentation, and has resolution equal to, or higher than, immunogold labelling.

# Materials and methods

**Key resources table**

| Reagent type (species) or resource | Designation | Source or reference | Identifiers | Additional information |
|---|---|---|---|---|
| Strain, strain background (*Leishmania tarentolae*) | LEXSY host P10 | Jena Biosciences | LT-101 | |
| Genetic reagent (*Mus musculus*) | Tpm3.1-APEX2 ± heterozygous | PMID:31331962 | | |
| Cell line (*Mesocricetus auratus*) | BHK-21 | ATCC | CCL-10 | |
| Cell line (*Homo sapiens*) | A-431 | ATCC | CRL-1555 | |
| Cell line (*Cricetulus griseus*) | Flp-In-CHO | Invitrogen | R75807 | |
| Transfected construct (synthetic) | LifeAct-APEX2 | This study | RRID:170523 | |
| Transfected construct (*Mus musculus*) | Cavin4-APEX | This study | RRID:170524 | |
| Transfected construct (synthetic) | pCSDEST2 | PMID:17948311 | RRID:22424 | |
| Transfected construct (synthetic) | p3E-APEX2 | PMID:29621251 | RRID:108894 | |
| Transfected construct (synthetic) | p3E-APEX2-P2A-mKate2 | PMID:26585296 | RRID:61671 | |
| Transfected construct (*Mus musculus*) | pME-CAV1 | This study | RRID:170527 | |
| Transfected construct (synthetic) | pME-LifeAct | PMID:32709891 | RRID:109545 | |
| Transfected construct (*Mus musculus*) | pME-Cavin4 | This study | RRID:170528 | |
| Transfected construct (*Homo sapiens*) | $A_1AR$-APEX2 | This study | RRID:170529 | |
| Antibody | Anti-tropomyosin three mouse monoclonal | Sigma-Aldrich | MABT1335 | (1:1000) |
| Antibody | Anti-alpha tubulin rabbit monoclonal | Abcam | Ab52866 | (1:3000) |
| Recombinant DNA reagent | GFP-CAV1-APEX (cell-free) | This study | RRID:170525 | |
| Recombinant DNA reagent | CAV1-APEX (cell-free) | This study | RRID:170526 | |

*Continued on next page*

*Continued*

| Reagent type (species) or resource | Designation | Source or reference | Identifiers | Additional information |
|---|---|---|---|---|
| Recombinant DNA reagent | pCellFree_G03 | PMID:25529348 | | |
| Recombinant DNA reagent | pCellFree_G03 | PMID:25529348 | | |
| Sequence-based reagent | Antisplice leader oligonucleotide | PMID:19648909 | | CAATAAAGTACAGAA ACTGATACTTATATAGCGTT |
| Commercial assay or kit | FluoroTect Green$_{Lys}$ in vitro Translation Labeling System | Promega | L5001 | |
| Commercial assay or kit | MycoAlert Mycoplasma Detection Kit | Lonza | LT07-418 | |
| Chemical compound, drug | 25% EM grade glutaraldehyde | Electron Microscopy Services | 16220 | |
| Chemical compound, drug | 16% EM grade paraformaldehyde | Electron Microscopy Services | 15710 | |
| Chemical compound, drug | Uranyl acetate | Electron Microscopy Services | 22400 | |
| Chemical compound, drug | Lead citrate | ProSciTech | C073 | |
| Chemical compound, drug | DAB | Sigma-Aldrich | D5905 | |
| Chemical compound, drug | Hydrogen peroxide solution | Sigma-Aldrich | H1009 | |
| Chemical compound, drug | Osmium tetroxide | ProSciTech | C010 | |
| Chemical compound, drug | LX 112 Embedding Kit | Ladd Research Industries | 21210 | |
| Chemical compound, drug | Horseradish peroxidase – 25 mg Type VI-A | Sigma-Aldrich | P6782 | |
| Chemical compound, drug | Silver nitrate | Sigma-Aldrich | 209139 | |
| Chemical compound, drug | Gum arabic | Electron Microscopy Services | 25574 | |
| Chemical compound, drug | Gold chloride | Electron Microscopy Services | 16583 | |
| Chemical compound, drug | Hexamethy lenetetramine | Sigma-Aldrich | 398160 | |
| Chemical compound, drug | Sodium tetraborate decahydrate (borax) | Sigma-Aldrich | B9876 | |
| Chemical compound, drug | Sodium thiosulphate | Sigma-Aldrich | 72049 | |
| Chemical compound, drug | Bactotryptone | Beckton Dickinson | 211699 | |
| Chemical compound, drug | Hemin chloride | MP Biomedicals | 0219402505 | |
| Chemical compound, drug | ATP | Chem-Impex | 00015 | |
| Chemical compound, drug | GTP | Chem-Impex | 00348 | |
| Chemical compound, drug | Spermidine | Sigma-Aldrich | 85558–5G | |

*Continued on next page*

*Continued*

| Reagent type (species) or resource | Designation | Source or reference | Identifiers | Additional information |
|---|---|---|---|---|
| Chemical compound, drug | DTT | Sigma-Aldrich | D0632-10G | |
| Chemical compound, drug | Cr phosphate | Chem-Impex | 00072 | |
| Chemical compound, drug | PEG 3350 | Hampton Research | HR2-527 | |
| Chemical compound, drug | Prot Inhib C | Roche Diagnostics | 11 873 580 001 | |
| Chemical compound, drug | CTP | Chem-Impex | 00095 | |
| Chemical compound, drug | UTP | Chem-Impex | 00311 | |
| Chemical compound, drug | T7 polymerase | In-house purification | N/A | |
| Chemical compound, drug | Cr phosphokinase | Sigma-Aldrich | C3755-35KU | |
| Chemical compound, drug | Alanine | Sigma-Aldrich | A7627 | |
| Chemical compound, drug | Arginine | Sigma-Aldrich | A5006 | |
| Chemical compound, drug | Asparagine | Sigma-Aldrich | A0884 | |
| Chemical compound, drug | Aspartic acid | Sigma-Aldrich | A9256 | |
| Chemical compound, drug | Cysteine | Sigma-Aldrich | C7352 | |
| Chemical compound, drug | Glutamic acid | Sigma-Aldrich | 49449 | |
| Chemical compound, drug | Glutamine | Sigma-Aldrich | G3126 | |
| Chemical compound, drug | Glycine | Sigma-Aldrich | G7126 | |
| Chemical compound, drug | Histidine | Sigma-Aldrich | H8000 | |
| Chemical compound, drug | Isoleucine | Sigma-Aldrich | I2752 | |
| Chemical compound, drug | Leucine | Sigma-Aldrich | L8912 | |
| Chemical compound, drug | Lysine | Sigma-Aldrich | L5626 | |
| Chemical compound, drug | Methionine | Sigma-Aldrich | M9625 | |
| Chemical compound, drug | Phenylalanine | Sigma-Aldrich | P2126 | |
| Chemical compound, drug | Proline | Sigma-Aldrich | P0380 | |
| Chemical compound, drug | Serine | Sigma-Aldrich | S4500 | |
| Chemical compound, drug | Threonine | Sigma-Aldrich | T8625 | |
| Chemical compound, drug | Tryptophan | Sigma-Aldrich | T0524 | |

*Continued*

| Reagent type (species) or resource | Designation | Source or reference | Identifiers | Additional information |
|---|---|---|---|---|
| Chemical compound, drug | Tyrosine | Sigma-Aldrich | T8566 | |
| Chemical compound, drug | Valine | Sigma-Aldrich | V0500 | |
| Chemical compound, drug | Penicillin-streptomycin | Life Technologies | 15070–063 | |
| Chemical compound, drug | Potassium acetate | Sigma-Aldrich | P1190 | |
| Chemical compound, drug | Magnesium acetate | Amresco | 0131–1 KG | |
| Chemical compound, drug | LR clonase II Plus | Invitrogen | 12538120 | |
| Chemical compound, drug | Hygromycin-B | Thermo Fisher/ Invitrogen | 10687010 | |
| Chemical compound, drug | DMEM | Life Technologies | 11995065 | |
| Chemical compound, drug | L-glutamine | Life Technologies | 25030081 | |
| Chemical compound, drug | Fetal bovine serum | Sigma-Aldrich | F9423 | |
| Software, algorithm | ImageJ/FIJI | PMID:22743772 | | https://imagej.nih.gov/ij/ |
| Software, algorithm | ImageJ/FIJI Weka Plugin | ImageJ developers | | https://imagej.net/ Trainable_Segmentation |
| Software, algorithm | iTEM | Olympus | | https://www.emsis.eu/home/ |
| Software, algorithm | AttoBright: LabView and GUI for data acquisition, Matlab code and GUI for data analysis | PMID:31827096 | | https://gambinsiereckilab. github.io/AttoBright/ |
| Software, algorithm | MAPs | Thermo Fisher | https://www.fei.com/ software/maps/#gsc.tab=0 | |

## Cell culture

BHK (ATCC), A431 (ATCC), MEF (as described in *Meiring et al., 2019*), and Flp-In-CHO (Invitrogen) cells were maintained in Dulbecco's modified eagle medium (DMEM), supplemented with L-glutamine, and 10% fetal bovine serum at 37°C with 5% $CO_2$. CHO cells stably expressing the human adenosine $A_1$ G-protein-coupled receptor ($A_1AR$) with C-terminally labelled APEX2 were generated as described previously (*Baltos et al., 2016*). Expression was maintained by addition of 500 µg/mL hygromycin-B to culture medium. All cells were subject to quarterly mycoplasma testing using the MycoAlert (Lonza) mycoplasma detection kit.

## Plasmid construction

CAV1-APEX2, CAV1-EGFP-APEX2, Cavin4-APEX2-P2A-mKate2, and LifeAct-APEX2-P2A-mKate2 were produced using the multisite gateway system (Invitrogen) by recombination of pME-Cavin4, pME-CAV1, pME-LifeAct, p3E-APEX2-P2A-mKate2, p3E-APEX2, and pCSDEST2. Full details and unique repository identifiers are given in key resources table. $A_1AR$-APEX2 was produced by gateway cloning, whereby human $A_1AR$ in pENTR/D/TOPO vector was recombined into pEF5/FRT/V5-DEST-APEX2. The resultant expression vector encoded the human $A_1AR$ with C-terminally tagged APEX2, adjoined by a glycine-serine-rich linker. GFP-CAV1 was produced by cloning into pCell-Free_G03 (*Gagoski et al., 2015*) from human ORFeome library described in *Škalamera et al., 2011*.

## Transfection

Cells were seeded into 35 mm tissue culture dishes ON, then transfected with Lipofectamine 3000 as per manufacturer's instructions. Cells were left for 24 hr, then fixed and processed for EM 24 hr later.

## EM eLine

SEM array tomography was carried out in an Electron Beam Lithography system, the Raith eLINE PLUS. The system is equipped with the dual detector (Inlens, SE) and a laser interferometric stage. Prior to the image acquisition, the scan field was calibrated at 25,000× with the laser stage. The displayed image was captured at 2 kV with 30 μm aperture (beam current of 30 pA), using the SE detector, at a working distance of 2.2 mm. The use of laser interferometric stages allows near-perfect stitching of scan fields by overlapping just five pixels at the edge.

## Focussed ion beam

FIB-SEM tomography of the sample was carried out on a FEI Scios DualBeam FIB-SEM system equipped with a 30 kV Ga + column and a Pt gas injection system (FEI). The sample was tilted to 52° so that its top surface was aligned parallel to the focal plane of the FIB. A protective layer of Pt was deposited onto the top surface directly above the volume of interest. A trench was milled at one end of the volume of interest using a 7 nA beam current. The purpose of the trench was to provide the SEM with an unobstructed view of the exposed cross-section and to allow for the escape of sputtered material. Fiducial marks were milled into both the top surface of the sample and the sidewall of the trench. The surface of the exposed face was planarized by milling using successively lower ion beam currents, down to 100 pA. The Auto Slice and View automation package (FEI) was used to sequentially mill away 10-nm-thick segments of material followed by SEM imaging of each newly exposed surface using the In-lens Trinity 'T1' BSE detector. An electron beam acceleration voltage of 2 kV and current of 50 pA was used. The stage remained stationary during the entire sequence so that the surface of each cross-section is tilted 38° from the SEM column and in-column detector. The respective fiducial marks were detected by the automation package prior to each milling and imaging step and used to correct for image drift of the electron and ion beam, respectively.

## Tomography

The 200-nm-thick sections were cut on a Leica UC6 ultramicrotome. Grids were assembled into an Autogrid (Thermo Fisher) and loaded onto a 200 kV Thermo Fisher Talos Arctica fitted with a Falcon 3EC (Thermo Fisher) camera operated in linear mode and at room temperature (RT). Bidirectional dual axis tilt series were acquired at 1° increments from −60° to +60° under the control of Tomography software (Thermo Fisher). Tilt series were reconstructed using weighted back-projection with IMOD.

## Serial block-face EM

Images were additionally collected using a VolumeScope serial block-face EM (SBEM; Thermo Fisher, Waltham, MA) equipped with a low-vac backscatter detector (VS-DBS; Thermo Fisher). Plastic embedded samples were scanned in low vacuum (10 Pa) with a landing beam energy of 2.0 kV and a current of 0.1 nA. Images were acquired using MAPS software (Thermo Fisher, Waltham, MA) at a pixel scale of 5.9–7 nm and pixel dwell of 3 μs.

## DAB treatment and APEX-Gold enhancement

DAB treatment and APEX-Gold enhancement were performed using a modification of the method of Sedmak et al., originally developed to enhance immunoperoxidase staining of tissues (*Sedmak et al., 2009*). Cells grown in 3 cm dishes were fixed with glutaraldehyde (2.5%) in PBS, then washed in PBS and then in cacodylate buffer, pH 7.35. Fixed cells were incubated in freshly prepared 0.05% DAB solution in cacodylate buffer for 10 min at RT followed by incubation with 0.05% DAB solution containing 0.01% $H_2O_2$ for 30 min at RT. The cells were then washed with cacodylate buffer and further fixed in 2.5% GA in cacodylate buffer for 1 hr at 4°C to stabilize the DAB reaction product. After washing with cacodylate buffer, cells were immediately processed for APEX-Gold enhancement.

Cells in dishes were washed in triple distilled water (H$_2$O) for 4 × 15 min to remove phosphates and reduce artefactual Ag/Au particle deposition. Cells were then blocked prior to silver enhancement with an aqueous solution containing 1% BSA and 20 mM glycine for 20 min. Dishes containing cells were prewarmed at 60°C for 10 min. An enhancement solution containing 3% hexamethylenetetramine (C$_6$H$_{12}$N$_4$) in H$_2$O, 5% silver nitrate (Ag NO$_3$) in H$_2$O, and 2.5% disodium tetraborate (Na$_2$B$_4$O$_7$10H$_2$O) in H$_2$O, mixed in a ratio of 20:1:2 was prewarmed and added to the cells, then incubated for 15 min at 60°C. After washing in H$_2$O (3 × 5 min), cells were incubated with 0.05% tetrachlorogold (III) acid trihydrate (AuHCl$_4$3H$_2$O) in H$_2$O for 5 min at RT, washed in H$_2$O, and incubated with 2.5% sodium thiosulphate for 4 min at RT. In some experiments, as indicated, the enhancement solution was mixed at a 1:1 ratio with an aqueous 50% gum arabic solution. These dishes were later rinsed with warmed H$_2$O to facilitate removal of residual gum arabic.

Cells were then postfixed with 1% osmium for 2 min, then serial dehydrated with increasing percentages of ethanol. Cells then underwent serial infiltration with LX112 resin in a Pelco Biowave, then incubated at 60°C for 24 hr. Ultrathin sections were attained on a ultramicrotome (UC6, Leica) and imaged using a JEOL1011 transmission EM at 80 kV. Where indicated sections were poststained with 2% aqueous uranyl acetate and Reynold's lead citrate.

### Freeze substitution

Cells grown on Thermanox coverslips were fixed with 4% PFA and 0.1% GA, then DAB-treated and underwent the APEX-Gold enhancement described earlier. Cells were then infiltrated with 2.1 M sucrose at RT for 1 hr, fast-frozen in liquid nitrogen, and freeze-substituted in 0.2% uranyl acetate in methanol, then washed in methanol and infiltrated with Lowicryl (HM20) resin before polymerizing at −50°C.

### Cryo-sectioning

Cells were fixed in 4% PFA and 0.1% glutaraldehyde and washed in PBS three times. Cells were scraped gently from the petri dish surface and pelleted at 8000 rpm in warm (37°C) 10% gelatine/PBS and then allowed to set on ice. The gelatine-embedded pellet was removed from the tube, then cut into small cubes and infused with 2.1 M sucrose at RT (three changes, 5 min each), mounted on aluminium stubs plunged into liquid nitrogen and then sectioned at −120°C on a Leica Ultra cut UC6 ultramicrotome. Ribbons of sections with a thickness of approximately 65–90 nm were placed on to 100 mesh formvar-coated Cu grids with a thin layer of carbon evaporated onto the surface. Grids complete with mounted sections were contrasted using a solution of 0.3% uranyl acetate and methyl cellulose on ice for 8 min. Grids were then picked up in wire loops and the excess methyl cellulose solution removed using filter paper.

### Light microscopy

Brightfield microscopy was carried out on a Zeiss 880 confocal microscope using the TPMT channel and 40× water immersion objective.

### Dot blots

Horseradish peroxidase or in vitro generated CAV1-GFP or CAV1-GFP-APEX2 were dotted onto nitrocellulose in a volume of 10 μL to give the indicated protein amounts. After drying, nitrocellulose was incubated with a blocking solution of 5% BSA. After washing with PBS, the nitrocellulose was incubated with the DAB solutions, before treatment with or without the APEX-Gold enhancement solutions.

### Western blotting of Tpm3.1 APEX2 cells

Triplicates of Tpm3.1-APEX2 +/− and −/− PMEFs were grown on 6 cm dishes until full confluency was reached. Cell lysates were harvested in 4°C RIPA buffer with protease inhibitor (cOmplete, EDTAfree Protease Inhibitor Cocktail, Merck) and homogenized by sonication for 30 s. Protein concentration was measured with Precision Red Assay (Cytoskeleton). Laemmli sample buffer (Biorad) was added 1:4 (v/v) and lysates were boiled at 95°C for 10 min. Samples were run at 100 V for 90 min on 10% polyacrylamide SDS-PAGE gels in running buffer. Gels were semi-dry-transferred in transfer buffer to PVDF membranes preactivated with 100% methanol. Membranes were blocked in

5% skim milk in TBS for 1 hr and probed with mouse γ/9d 2G10.2 (1:1000, MERCK MABT1335), rabbit anti-mouse IgG (1:3000, Abcam ab97046), rabbit α-tubulin (1:3000) (Abcam ab52866) and goat anti-rabbit IgG (1:5000) (Biorad 170-6515) antibodies sequentially for 1 hr. Luminata Crescendo Western HRP substrate (Merk) was used for imaging on a Chemicoc MP imaging system (Biorad). Band densitometry was quantified (ImageJ) and normalized to α-tubulin control.

## Cell-free expression and particle characterization, cell incubation

*Leishmania tarentolae* cell-free lysate was produced, and cell-free protein expression was performed as described by *Hunter et al., 2018*. Briefly, *Leishmania tarentolae* Parrot strain was obtained as LEXSY host P10 from Jena Bioscience GmbH, Jena, Germany, and cultured in TBGG medium containing 0.2% v/v penicillin/streptomycin (Life Technologies) and 0.05% w/v hemin (MP Biomedical). Cells were harvested by centrifugation at $2500 \times g$, washed twice by resuspension in 45 mM HEPES, pH 7.6, containing 250 mM sucrose, 100 mM potassium acetate, and 3 mM magnesium acetate and resuspended to 0.25 g cells/g suspension. Cells were placed in a cell disruption vessel (Parr Instruments, Moline, IL) and incubated under 7000 KPa nitrogen for 45 min, then lysed by rapid release of pressure. The lysate was clarified by sequential centrifugation at $10,000 \times g$ and $30,000 \times g$ and anti-splice leader oligonucleotide was added to 10 µM. The lysate was then desalted into 45 mM HEPES, pH 7.6, containing, 100 mM potassium acetate and 3 mM magnesium acetate and snap-frozen until required.

Cell-free lysate was supplemented with a feeding solution containing nucleotides, amino acids, T7 polymerase, HEPES buffer, and a creatine/creatine kinase ATP regeneration system at a ratio of lysate to feed solution of 0.21 and a final $Mg^{2+}$ concentration of 6 mM. Purified plasmid DNA, at a concentration of 1000 ng/µL, was added to the expression reaction at a ratio of 1:9 (v/v), and the reaction allowed to proceed for 3 hr at 27°C. When visualization of non-GFP-tagged expressed protein was desired, the FluoroTect GreenLys in vitro Translation Labeling System (Promega) was used: labelled tRNA was diluted 1:10 from the supplied material and added to expression reactions at a ratio of 1:9 (v/v). Fluorescently tagged/labelled expressed protein was detected both before and after SDS-PAGE using a Chemidoc MP imaging system (Bio-Rad, Laboratories Pty. Ltd., Gladesville, NSW, Australia) as described in *Hunter et al., 2018*.

Cell-free reaction product was diluted (1:1) with DMEM and incubated with cells for 5 min at 37°C in 5% $CO_2$. Cells were then fixed and processed for DAB treatment and APEX-Gold enhancement as stated above. Thin sections were imaged at a magnification of 120,000×. APEX-Gold positive vesicles of 50–100 nm were selected for quantitation of gold particles per vesicle (see, e.g., *Figure 2—figure supplement 1D'*).

## FCS method

Single-molecule fluorescence counting methods were used to compare the oligomeric state of the expressed CAV1-GFP-APEX2 construct with the known state of GFP-labelled CAV1 (GFP-CAV1). Comparison of the brightness values obtained for the two constructs suggests that vesicles produced using the CAV1-GFP-APEX2 construct comprised approximately 110 GFP-CAV1-APEX2 molecules.

Single-molecule spectroscopy was performed using an AttoBright instrument optimized for the detection of GFP (*Brown et al., 2019*). A 488 nm laser was focussed into the sample solution using a C-Apochromat 40×/1.2 W water immersion objective lens (Zeiss) and the fluorescence emission was filtered using a 500–550 nm bandpass filter. Samples were diluted 1:5 directly from cell-free expression with buffer EB and placed in a custom-made silicone 192-well plate with a $70 \times 80$ mm² glass coverslip (ProSciTech). For single-molecule burst brightness analysis, the frequency of events for each range of GFP fluorescence intensity was counted and plotted on a histogram.

## Acknowledgements

The authors acknowledge the help of staff and use of facilities in the Microscopy Australia NCRIS Facility at the Centre for Microscopy and Microanalysis at The University of Queensland. The authors acknowledge the use of the Cryo Electron Microscopy Facility through the Victor Chang Innovation Centre, funded by the NSW government, the Electron Microscope Unit within the Mark Wainwright Analytical Centre at UNSW Sydney and Microscopy Australia. This work was supported by the

National Health and Medical Research Council of Australia (grants APP1140064 and APP1150083 and fellowship APP1156489 to RGP; APP1185021 to NA). RGP is supported by the Australian Research Council (ARC) Centre of Excellence in Convergent Bio-Nano Science and Technology. This work was also supported by an Australian Department of Industry, Innovation and Science Cooperative Research Centre Project (CRC-P) grant to PWG and ECH, and grants from the Australian Research Council (ARC grant DP160101623), the Australian National Health and Medical Research Council (NHMRC grant APP1100202, APP1079866), and The Kid's Cancer Project to PWG and ECH. MES is funded by an International Cotutelle Macquarie University Research Excellence Scholarship (iMQRES 2019060).

## Additional information

### Funding

| Funder | Grant reference number | Author |
|---|---|---|
| National Health and Medical Research Council | APP1140064 | Robert G Parton |
| National Health and Medical Research Council | APP1150083 | Arthur Christopoulos Robert G Parton |
| National Health and Medical Research Council | APP1156489 | Robert G Parton |
| National Health and Medical Research Council | APP1185021 | Nicholas Ariotti |
| Australian Research Council | CE140100036 | Robert G Parton |
| Department of Industry, Innovation and Science, Australian Government | CRC-P | Edna C Hardeman Peter W Gunning |
| Australian Research Council | DP160101623 | Edna C Hardeman Peter W Gunning |
| National Health and Medical Research Council | APP1100202 | Edna C Hardeman Peter W Gunning |
| National Health and Medical Research Council | APP1079866 | Edna C Hardeman Peter W Gunning |
| The Kid's Cancer Project | | Edna C Hardeman Peter W Gunning |
| Macquarie University | iMQRES 2019060 | Marcel Ethan Sayre |

The funders had no role in study design, data collection and interpretation, or the decision to submit the work for publication.

### Author contributions

James Rae, Data curation, Formal analysis, Validation, Investigation, Visualization, Methodology, Writing - original draft, Writing - review and editing; Charles Ferguson, Validation, Investigation, Methodology; Nicholas Ariotti, Formal analysis, Investigation, Visualization, Writing - original draft, Writing - review and editing; Richard I Webb, Han-Hao Cheng, James L Mead, James D Riches, Dominic JB Hunter, Nick Martel, Joanne Baltos, Sachini Fonseka, Marcel E Sayre, Thomas E Hall, Investigation; Arthur Christopoulos, Nicole S Bryce, Maria Lastra Cagigas, Edna C Hardeman, Peter W Gunning, Yann Gambin, Resources, Investigation; Robert G Parton, Conceptualization, Resources, Data curation, Formal analysis, Supervision, Funding acquisition, Visualization, Methodology, Writing - original draft, Project administration, Writing - review and editing

### Author ORCIDs

James D Riches http://orcid.org/0000-0001-8494-4743
Dominic JB Hunter http://orcid.org/0000-0002-1826-6902
Nicole S Bryce https://orcid.org/0000-0001-9799-7393

Edna C Hardeman [ID] http://orcid.org/0000-0003-1649-7712
Peter W Gunning [ID] http://orcid.org/0000-0003-0833-3128
Yann Gambin [ID] http://orcid.org/0000-0001-7378-8976
Thomas E Hall [ID] http://orcid.org/0000-0002-7718-7614
Robert G Parton [ID] https://orcid.org/0000-0002-7494-5248

## Decision letter and Author response

Decision letter https://doi.org/10.7554/eLife.64630.sa1
Author response https://doi.org/10.7554/eLife.64630.sa2

## Additional files

### Supplementary files

• Transparent reporting form

### Data availability

All data generated or analysed during this study are included in the manuscript.

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
