## [Decision Letter]

**Acceptance summary:**

The ability to label proteins of interest for visualization by microscopy revolutionized biology. Prior work developed the label known as APEX, an ascorbate peroxidase enzyme, for ultrastructural localization by electron microscopy. APEX labeling depends on oxidizing a small molecule, diaminobenzidine, but the resulting contrast is sometimes too weak or diffuse for quantitative labeling or detection of low-abundance proteins. Here, Rae et al. have developed an extension of APEX labeling, named APEX-Gold, which harnesses the oxidized diaminobenzidine reaction product of APEX to generate silver/gold heavy atom particles near the labeled protein. APEX-Gold labeling, therefore, has increased signal-to-noise and is suitable for quantitative studies of even low-abundance proteins, and is compatible with a diverse array of electron microscopy methods.

**Decision letter after peer review:**

Thank you for submitting your article "APEX-Gold: A genetically-encoded particulate marker for robust 3D electron microscopy" for consideration by *eLife*. Your article has been reviewed by 3 peer reviewers, one of whom is a member of our Board of Reviewing Editors, and the evaluation has been overseen by Anna Akhmanova as the Senior Editor. The following individuals involved in the review of your submission have agreed to reveal their identity: Yannick Schwab (Reviewer #2); Paul Verkade (Reviewer #3).

The reviewers have discussed the reviews with one another and the Reviewing Editor has drafted this decision to help you prepare a revised submission.

Summary:

The manuscript by Rae et al. reports a new protocol for labeling genetically-tagged proteins of interest with heavy atom particles for visualization by electron microscopy. The optimized protocol builds on the use of the enzyme APEX2, fused to the target protein of interest. The contrast enhancement may be useful in diverse 3D EM techniques. Also, reviewers were enthusiastic about the prospects for quantitative studies, even for low-levels of endogenous expression, because the new method appears to improve the proportionality of the signal such that the number of APEX2 tags in a sample correlates with the number of heavy atom particles. The apparent simplicity of the protocol raises the potential for it to become a standard in the field of EM labeling.

Essential revisions:

1. The technique is to improve the contrast of APEX2 labeling, yet, it seems that on most of the EM images, post-staining was not performed? Could the authors comment, or also show images from post-stained sections? This is particularly obvious on the supplementary Fig6a, which is supposed to also display a non-transfected cell. In fact, this cell is almost invisible due to the absence of contrast.

2. The experiments on freeze-substituted material show staining that does not appear associated with caveola. Has this been analyzed? Could this be a limitation of the workflow? We understand the strong interest in adapting the technique to post-embedding, on-section immuno-labeling, but this would only be valid if the initial staining was not affected. Please clarify the methods section for this part of the manuscript as no information is available on the initial fixation (prior to FS). Are the cells also fixed with high levels of glutaraldehyde? Or are they mildly fixed to enable better success for on-section immunolabelling?

3. We do not fully understand the take-home-message from the last set of experiments, intending to set a labeling standard. Is the intention to prove the labeling can be quantitative? Have the authors tried to relate the number of particles to the number of APEX2 fusion proteins expected to be expressed at these vesicles (as suggested by the FCS experiment)? This might require further experiments or explanations. Overall, the issue of quantifying size distributions needs more explication and discussion. What is the detection limit? Can image analysis tools automate identification and measurement?

---

## [Author Response]

Essential revisions:1. The technique is to improve the contrast of APEX2 labeling, yet, it seems that on most of the EM images, post-staining was not performed? Could the authors comment, or also show images from post-stained sections? This is particularly obvious on the supplementary Fig6a, which is supposed to also display a non-transfected cell. In fact, this cell is almost invisible due to the absence of contrast.

This is an important point as one advantage of the APEX-gold technique is the opportunity to increase the staining of the cellular structure because of the greater signal to noise of the gold particles as compared to the standard DAB reaction product. Routinely we keep staining to a minimum to make sure all gold particles are visualized, but we have now provided a new image in the revised Figure 1—figure supplement 2A to show the morphology after on-grid staining. We have changed the text to emphasize this point on page 4 as follows:

“Sections were generally viewed without further on-grid staining to maximize detection of gold particles. However, visualization of the electron dense APEX-Gold particles is readily compatible with on-grid staining (Figure 1-supplement figure 2A).”

2. The experiments on freeze-substituted material show staining that does not appear associated with caveola. Has this been analyzed? Could this be a limitation of the workflow? We understand the strong interest in adapting the technique to post-embedding, on-section immuno-labeling, but this would only be valid if the initial staining was not affected.

The reviewer raised the possibility of a redistribution of the deposited gold particles. We were also puzzled by this and so we tested the use of a more rapid processing scheme including a 1h cryoprotectant infiltration and embedding at low temperature with HM20 resin. As shown in the revised figure (Figure 1—figure supplement 4D,E), this newly optimised protocol yields gold particles that are clearly visible and well-retained after the freezing, freeze substitution, and low temperature-embedding in HM20. We provide these images as a proof-of-principle demonstration that the technique is compatible with these methods and can be compatible with on-section labelling methods. This is now described on page 4 as follows:

“We also tested the compatibility of the method with freeze substitution/low temperature embedding (Figure 1-supplement figure 4D,E). This makes APEX-Gold potentially compatible with double labeling, with the APEX fusion protein being expressed endogenously in the cells and then sections labeled for other proteins of interest.”

Please clarify the methods section for this part of the manuscript as no information is available on the initial fixation (prior to FS). Are the cells also fixed with high levels of glutaraldehyde? Or are they mildly fixed to enable better success for on-section immunolabelling?

We thank the reviewer for this comment and apologise for the oversight. In fact, we did employ a mild fixation of 4% pfa/0.1% glutaraldehyde prior to cryoprotection and freezing. This is now described on page 14. The relevant passage reads:

“**Cells grown on Thermanox coverslips were fixed with 4% PFA and 0.1% GA then DAB treated and underwent the APEX-Gold enhancement described earlier.”**

3. We do not fully understand the take-home-message from the last set of experiments, intending to set a labeling standard. Is the intention to prove the labeling can be quantitative? Have the authors tried to relate the number of particles to the number of APEX2 fusion proteins expected to be expressed at these vesicles (as suggested by the FCS experiment)? This might require further experiments or explanations. Overall, the issue of quantifying size distributions needs more explication and discussion. What is the detection limit? Can image analysis tools automate identification and measurement?

We have now rewritten this section with further explanation. There were two aims for the experiments with CAV1-APEX2 vesicles. The first was to make a preparation of APEX particles that can act as a positive control for the method by use of a simple dot blot; This can be run alongside the experiment to ensure that the enhancement process works. This is a simple but very important step for the successful adoption of a new method by different laboratories (one problem with enhancement methods is that a failed experiment is only revealed after processing, sectioning, and then viewing in the electron microscope). We now stress this in the revised manuscript.

Secondly, the CAV1-APEX vesicles have a defined number of APEX molecules and so can potentially be used as a standard for quantitation. In the revised manuscript we now provide the quantitation of APEX-gold per vesicle (page 5) and provide images showing how the counting was performed (revised Figure 2—figure supplement 1D). We outline the existing challenges with the method (including ‘fused gold particles’ and the possibility that the enhancement might vary in different compartments) in the methodological consideration (previously entitled ‘Limitations’) section. Most importantly, we hope that these experiments stimulate the use of standards to add more quantitation to APEX-based localization studies and we now make this point more clearly in the Discussion.

“As shown in Figure 2B the GFP-CAV1-APEX2 vesicles were observed on the surface of the cells and in endosomes and are clearly decorated with the APEX-Gold enhanced particles (Figure 2—figure supplement 1D). […] This proof of principle experiment using exogenously added Caveolin-APEX vesicles as a standard demonstrates the potential of the APEX-Gold method for determination of the density of unknown proteins.”

On Page 7 we provide a description and discussion of this approach for application to molecular quantitation of electron microscopy data:

“By combining with an APEX-fusion protein of known signal density (here demonstrated by cell-free generated vesicle of a defined APEX density as a calibration standard) it is possible to quantitatively compare particle density to antigen density. While we demonstrated this principle by using exogenously added cell-free synthesized caveolin-AP vesicles, an APEX-tagged antigen generated by CRISPR technology that is present at known density in a specific region of the cell could also be used as a standard.”

We further outlined some of the caveats in using this approach in the final section entitled ‘Methodological Considerations’. The relevant passage now reads:

“Increased enhancement times can also cause more variable particle size including the appearance of ‘fused’ gold particles where individual APEX molecules are closely packed and so not clearly resolved as individual gold particles after enhancement (see for example Figure 2—figure supplement 1D). […] Optimisation of the enhancement time is required to allow detection of individual gold particles without formation of single large particles from multiple closely-packed APEX molecules.”

Our experiments detecting APEX-tagged tropomyosin from genetically-modified mice demonstrate the sensitivity of the technique, as described in the Discussion. While we can state that the method can detect an endogenous protein expressed at 5% of endogenous levels we cannot provide any further estimate of sensitivity as this will depend on the optimisation of the enhancement conditions for particular targets of interest. This experiment emphasizes the power of the technique when the APEX-tagged protein is expressed at levels below endogenous levels (ie. with gold particles well separated) and its compatibility with image analysis tools for detection, as used in Figure 2—figure supplement 2D,D’.